# Multi-Omics Approach Profiling Metabolic Remodeling in Early Systolic Dysfunction and in Overt Systolic Heart Failure

**DOI:** 10.3390/ijms23010235

**Published:** 2021-12-26

**Authors:** Antoine H. Chaanine, LeeAnn Higgins, Todd Markowski, Jarrod Harman, Maureen Kachman, Charles Burant, L. Gabriel Navar, David Busija, Patrice Delafontaine

**Affiliations:** 1Department of Medicine, Tulane University, New Orleans, LA 70112, USA; pdelafon@tulane.edu; 2Department of Physiology, Tulane University, New Orleans, LA 70112, USA; navar@tulane.edu; 3Department of Biochemistry, Molecular Biology, and Biophysics, University of Minnesota Medical School, Minneapolis, MN 55455, USA; higgi022@umn.edu (L.H.); marko025@umn.edu (T.M.); 4Department of Ophthalmology, Louisiana State University Health Sciences Center, New Orleans, LA 70112, USA; jharma@lsuhsc.edu; 5Michigan Regional Comprehensive Metabolomics Resource Core, University of Michigan, Ann Arbor, MI 48109, USA; mkachman@med.umich.edu; 6Department of Internal Medicine, University of Michigan Medical School, Ann Arbor, MI 48109, USA; burantc@med.umich.edu; 7Department of Pharmacology, Tulane University, New Orleans, LA 70112, USA; dbusija@tulane.edu; 8Tulane Brain Institute, Tulane University, New Orleans, LA 70112, USA

**Keywords:** heart failure, metabolic remodeling, mitochondria, PKA, AMPK, calcium cycling

## Abstract

Metabolic remodeling plays an important role in the pathophysiology of heart failure (HF). We sought to characterize metabolic remodeling and implicated signaling pathways in two rat models of early systolic dysfunction (MOD), and overt systolic HF (SHF). Tandem mass tag-labeled shotgun proteomics, phospho-(p)-proteomics, and non-targeted metabolomics analyses were performed in left ventricular myocardium tissue from Sham, MOD, and SHF using liquid chromatography–mass spectrometry, *n* = 3 biological samples per group. Mitochondrial proteins were predominantly down-regulated in MOD (125) and SHF (328) vs. Sham. Of these, 82% (103/125) and 66% (218/328) were involved in metabolism and respiration. Oxidative phosphorylation, mitochondrial fatty acid β-oxidation, Krebs cycle, branched-chain amino acids, and amino acid (glutamine and tryptophan) degradation were highly enriched metabolic pathways that decreased in SHF > MOD. Glycogen and glucose degradation increased predominantly in MOD, whereas glycolysis and pyruvate metabolism decreased predominantly in SHF. PKA signaling at the endoplasmic reticulum–mt interface was attenuated in MOD, whereas overall PKA and AMPK cellular signaling were attenuated in SHF vs. Sham. In conclusion, metabolic remodeling plays an important role in myocardial remodeling. PKA and AMPK signaling crosstalk governs metabolic remodeling in progression to SHF.

## 1. Introduction

The heart is an organ with high-energy demands, which has the flexibility to utilize almost any type of substrate for adenosine triphosphate (ATP) production [1]. It cycles nearly 6 kg of ATP per day, produced in the mitochondria by oxidative phosphorylation (OXPHOS). Ninety-five percent of the generated ATP is consumed by myosin ATPase, the sodium/potassium pump, and the sarco/endoplasmic reticulum calcium ATPase (Serca2a or Atp2a2), which is necessary for the contraction and maintenance of ion homeostasis during the contraction and relaxation phases of the cardiac cycle, respectively [2]. Mitochondria are abundant in the heart and constitute about 30% of the cardiomyocyte volume [3,4]. In addition to their role in OXPHOS and ATP production, mitochondria play a role in myocardial calcium and redox signaling, ion homeostasis, and metabolism. This multiplicity of mitochondrial (mt) function is tightly coordinated and balanced to maintain cell survival [5].

Perturbations in mt function and metabolic remodeling are evident in heart failure (HF) [6,7,8]. Changes in mt morphology and dynamics take place early in pathological hypertrophy and worsen during transition to advanced systolic HF (SHF) [9,10]. Moreover, fatty acid (FA) utilization, mt FA β-oxidation, and OXPHOS are impaired early in pathological hypertrophy and gradually decline with the progression of myocardial remodeling and systolic dysfunction [6,11]. The heart initially compensates through increases in glucose uptake and glycolysis, but this is eventually impaired upon progression to SHF [6,11]. Less is known about when changes in amino acid (AA) and nucleotide metabolism and branched-chain amino acid (BCAA) catabolism occur and their contribution to myocardial remodeling. Derangements in cardiac metabolism are debated as to whether they are the cause or a marker of progression to SHF. Evidence suggests that the former is true. Changes in myocardial signaling as a result of altered workload and biomechanical and neurohumoral stress are evident in the earliest phases of cardiac stress and drive the initiation of cardiac remodeling and its progression toward SHF [12]. Kinases and phosphatases, such as protein kinase A (PKA), calcium/calmodulin-dependent protein kinase II (subunits delta (CaMKIIδ), and gamma (CaMKIIγ)), and calcineurin that drive cardiac remodeling and altered calcium signaling [13,14,15] are also implicated in metabolic reprogramming [16,17,18,19]. Moreover, classic metabolites and metabolic intermediates, summarized elsewhere [1], have been recognized as modulators of signal transduction and ion channels/calcium homeostasis, suggesting that metabolic remodeling is an important player in myocardial remodeling and the progression to overt SHF.

What remains unknown is: (1) a global understanding of metabolic disturbances, primarily changes in mitochondrial proteome and phosphoproteome in early systolic dysfunction and in transition to overt SHF, and (2) the time point at which derangements in signal transduction pathways, primarily PKA and adenosine monophosphate-activated protein kinase (AMPK) signaling modulating metabolic remodeling occur, and which of those predominantly drive metabolic remodeling in the transition to overt SHF. To gain a global understanding of metabolic remodeling at the proteome and metabolome level, and implicated signal transduction pathways, we performed tandem mass tag (TMT)-labeled shotgun proteomics, untargeted metabolomics, and shotgun phospho (p)-proteomics analysis of left ventricular (LV) myocardium tissue in two rat models of induced pressure overload (PO): (1) moderate remodeling and early systolic dysfunction (MOD), and (2) overt SHF.

## 2. Results

The echocardiographic parameters of the studied animals are presented in Appendix A. At week 3 post ascending aortic banding (AAB), there was evidence of concentric hypertrophy in the MOD and SHF groups with increases in LV septal and posterior wall thicknesses, and LV ejection fraction (LVEF) compared with Sham. At week 8 post-AAB, the MOD and SHF phenotypes continued to show the same degree of concentric hypertrophy, but developed increases in LV end-diastolic (LVEDV) and end-systolic volumes (LVESV) and a decrease in LVEF compared with week 3 post-AAB. Moreover, the SHF phenotype showed significantly higher LVEDV and LVESV, but lower LVEF than the MOD phenotype. The heart to body weight (BW) ratio increased in SHF > MOD vs. Sham. The LV weight to BW ratio increased equally in both MOD and SHF vs. Sham; however, right ventricular weight to BW ratio increased in SHF compared with MOD and Sham.

### 2.1. Visualization of the MOD and SHF Proteomic Datasets

Heat maps and PCA plots (Appendix A) show the differential log2 fold change and the variance in biological samples, respectively, for the total identified proteins in Sham and MOD and Sham and SHF proteomic runs. The PCA plots show significant separation of the Sham and MOD, and Sham and SHF biological samples, suggesting a phenotypic change in MOD and SHF groups compared with Sham. Heat maps, volcano plots, and PCA plots (Figure 1A,B and Appendix A), respectively, show the differential log2 fold change, the log2 fold change in group means, and the variance in biological samples, respectively, for the identified proteins that changed in MOD and SHF vs. Sham. A total of 227 and 722 proteins were down-regulated in MOD and SHF, respectively, vs. Sham. Of these, 55% (125/227) and 45% (328/722) (Figure 1C), respectively, were mt proteins (intersection of green with blue diagrams and yellow with red diagrams, respectively). A total of 318 and 783 proteins were up-regulated in MOD and SHF vs. Sham, respectively (Figure 1C). Of these, 3% (9/318) and 1.3% (10/783), respectively, were mt proteins. The corresponding analysis in Scaffold is presented in Appendix A. Heat maps, volcano plots, and PCA plots of the identified mt proteins that changed in MOD and SHF vs. Sham, analyzed in Qlucore, are presented in Appendix A. The corresponding analysis in Scaffold, including protein–protein interaction network by confidence, is presented in Appendix A. Transitioning from MOD to SHF was associated with an increase in the number of down-regulated mt proteins vs. Sham (32% (125/393) vs. 67% (328/49)). One hundred and fourteen mt proteins were downregulated in MOD and SHF vs. Sham. An additional 11 mt proteins were down-regulated in MOD vs. Sham; whereas an additional 214 (160 + 54) mt proteins were down-regulated in SHF vs. Sham (Figure 1C). A total of 160 of these were identified in the MOD proteomic run (but did not reach statistical significance), and 54 were not. A similar trend was seen in the dataset analyzed by Scaffold. Volcano plots (Appendix A), show the log2 fold change in group means for the 114 and 160 mt proteins in MOD and SHF vs. Sham that were discussed above.

Heat maps, volcano plots, and PCA plots (Figure 1D,E as well as Appendix A) show the differential log2 fold change and the log2 fold change in group means, as well as the variance in biological samples, respectively, for the identified proteins related to metabolism and respiration that changed in MOD and SHF vs. Sham. Transitioning from MOD to SHF was associated with an increase in the number of metabolism- and respiration-related proteins that were predominantly down-regulated, 125 and 306, in MOD and SHF, respectively, vs. Sham (Figure 1F); 82% (103/125) and 71% (218/306) of these were mt proteins. Corresponding analysis in Scaffold is presented in Appendix A. Volcano plots (Appendix A) show the log2 fold change in group means for the 113 (95 + 18) commonly identified proteins related to metabolism and respiration that were down-regulated in MOD and SHF vs. Sham, and for the 100 commonly identified mt proteins involved in metabolism and respiration that were down-regulated in SHF only. Data visualization of the identified mt proteins, not involved in metabolism or respiration, that changed in MOD and SHF vs. Sham are presented in Appendix A.

Heat maps (Figure 2A and Appendix A) show the canonical metabolic pathways enriched by z-score and *p*-value in the MOD and SHF vs. Sham. OXPHOS followed by mt FA β-oxidation were the highest enriched and inhibited metabolic pathways (by *p*-value and z-score, respectively) in SHF > MOD vs. Sham. BCAA catabolism, TCA cycle, glutamine, and tryptophan degradation were also highly enriched and inhibited, by *p*-value and z-score, respectively, in SHF > MOD vs. Sham. Glycogen degradation was mainly activated in MOD, whereas glucose biosynthesis and UDP-N-acetyl-D-galactosamine biosynthesis were activated in MOD > SHF vs. Sham. Glycolysis was enriched in SHF > MOD and inhibited in MOD and SHF; however, the z-score was below the cutoff value of 2. Acetate conversion to acetyl-coenzyme A (CoA) and acetyl-CoA biosynthesis were enriched and inhibited in SHF vs. MOD and Sham. Purine, guanosine, and adenosine nucleotide degradation were enriched and activated in SHF > MOD; however, the z-score was below the cutoff value of 2. Ketolysis was enriched in SHF > MOD, and was inhibited in both MOD and SHF, but the z-score was below the cutoff value of 2.

Of the upstream regulators (Figure 2A) that were highly enriched in MOD and SHF vs. Sham, the peroxisome proliferator-activated receptor gamma coactivator 1-alpha (PPARGC1A) and the peroxisome proliferator-activated receptor alpha (PPARA) were inhibited in SHF > MOD vs. Sham, whereas the mitogen-activated protein kinase kinase 4 (MAP4K4), which is upstream of Jun N-terminal kinase (JNK), and the transforming G growth factor beta 1 (TGFB1) were activated in SHF > MOD vs. Sham. The PPARGC1A and carnitine palmitoyltransferase 1B (CPT1B) gene networks in MOD and SHF vs. Sham are presented in Appendix A, respectively. PGC-1α expression (Figure 2B) was decreased in MOD vs. Sham and further decreased in SHF vs. MOD and Sham. The expression of the mt trifunctional enzymes (HADHA and HADHB) and the mt medium-chain acyl-CoA dehydrogenase (MCAD) proteins, involved in mt FA β-oxidation, were decreased in MOD and SHF vs. Sham (Figure 2B). HADHB expression was further decreased in SHF vs. MOD and Sham. The OXPHOS network of the electron transport chain (ETC) complexes I-IV and ATP synthase (complex V) (Figure 3A) show that the ETC complexes II, IV and ATP synthase were decreased/inhibited in MOD vs. Sham, whereas all the ETC complexes I-IV and ATP synthase were down-regulated/inhibited in SHF vs. Sham. ETC complexes I and III were somewhat affected in MOD vs. Sham. ETC complexes III, IV, and ATP synthase expression were decreased in MOD vs. Sham and further decreased in SHF vs. MOD and Sham (Figure 3B). ETC complex II expression was equally decreased in MOD and SHF, whereas ETC complex I expression was decreased in SHF only (Figure 3B).

### 2.2. Visualization of the Untargeted Metabolomics Dataset in Sham, MOD and SHF Groups

Heat maps, volcano plots, and PCA plots (Figure 4A–C) show the differential log2 fold change, the log2 fold change in group means, and the variance in biological samples, respectively, for the identified metabolites that changed in MOD and SHF vs. Sham. The PCA plots indicate that the MOD and SHF biological samples share a similar unidirectional change for the metabolites that changed in MOD vs. Sham, whereas the biological samples in the MOD group are centered midway between the Sham and SHF groups for the metabolites that changed in SHF vs. Sham. These data suggest a phenotypic disease progression upon transitioning from MOD to an advanced SHF phenotype, similar to that shown by the proteomic datasets. A total of 45 metabolites changed in MOD (blue) and 173 in SHF (red) vs. Sham and 56 (green) changed in SHF vs. MOD (Figure 4D). Eighteen common metabolites changed in MOD and SHF vs. Sham (intersection of red and blue diagrams). Seven of these also changed in SHF vs. MOD (intersection of all three diagrams). Of the 11 commonly identified metabolites that changed in MOD and SHF vs. Sham, there was a trend toward the further down-regulation of 2′,3′ cyclic AMP (or 3′,5′ cyclic AMP) and pantetheine 4′-phosphate in SHF compared to MOD (Figure 4E). Heat maps and PCA plots of the 18 commonly identified metabolites that changed in MOD and SHF vs. Sham and of the 56 metabolites that changed in SHF vs. MOD and Sham are presented in Figure 4F,G, respectively. Figure 5A,B are a heat map and PCA plot of the fatty acyl-carnitines that changed in abundance in MOD and SHF vs. Sham. The very long- and long-chain acyl-carnitines were equally down-regulated in MOD and SHF vs. Sham, whereas the medium- and short-chain acyl-carnitines decreased in abundance in SHF vs. Sham and down-trended in MOD vs. Sham.

Heat maps, generated by Ingenuity Pathway (IPA) comparison analysis (Figure 5C), show the metabolic pathways and upstream regulators that were enriched by *p*-value and z-score, respectively, in MOD and SHF vs. Sham. Metabolic pathways related to CMP-N-acetylneuraminate biosynthesis, glycolysis, taurine biosynthesis, valine degradation, glutamine, and sucrose degradation were enriched in MOD vs. Sham. However, CMP-N-acetylneuraminate biosynthesis, glycolysis, isoleucine, leucine and valine degradation, UDP-N- acetyl-D-glucosamine biosynthesis, glutamine degradation, and citrulline metabolism metabolic pathways were enriched in SHF vs. Sham. Bar graphs (Appendix A) show the metabolic pathways that were expected to be activated (orange) or inhibited (blue) in MOD and SHF vs. Sham and in SHF vs. MOD. A z-score could not be estimated due to the limited number of identified metabolites per pathway or lack of information on the expected change in certain metabolites per metabolic pathway in IPA. Transitioning from MOD to SHF was associated with the down-regulation of glycolysis, BCAA catabolism, and gluconeogenesis, as well as the up-regulation of taurine biosynthesis, CMP-N-acetylneuraminate biosynthesis, and methionine degradation. Enrichment pathway analysis (Appendix A) shows the enrichment ratio for the identified metabolites that changed in MOD and SHF vs. Sham. Of the upstream regulators that were highly enriched, calcium and calcium-independent phospholipase A2 (PNPLA8) were decreased/inhibited in SHF vs. Sham only (Figure 5C). The calcium mechanistic network (Figure 5D) shows the associated metabolites involved in calcium cycling (2′,3′ cyclic AMP) and calcium homeostasis regulation (eicosanoids) that were down-regulated in SHF only vs. Sham.

### 2.3. Visualization of the MOD and SHF p-Proteomic Datasets

A total of 817 and 961 proteins, as well as 2034 and 2524 p-sites were identified in MOD and SHF p-proteomic runs, respectively (Appendix A). Venn diagrams (Appendix A) show the number of p-sites that changed in MOD and SHF vs. Sham, analyzed in Scaffold PTM and Qlucore. Heat maps, volcano plots, and PCA plots (Figure 6A–C) show the differential log2 fold change, the log2 fold change in group means, and the variance in biological samples, respectively, for the identified p-sites that changed in MOD and SHF vs. Sham. One of the Sham biological samples in the MOD p-proteomic run was an outlier and was eliminated because it skewed the data (heat map and PCA plot: Appendix A). Corresponding analysis in Scaffold PTM is presented in Appendix A. Heat maps, generated by IPA comparison analysis (Figure 7A and Appendix A), show signaling pathways that were enriched by p-value and z-score in MOD and SHF vs. Sham. PKA and AMPK signaling, two important signaling pathways regulating metabolism, were enriched in SHF > MOD vs. Sham, but were inactivated in SHF only vs. Sham. These results were validated by Western blotting (Figure 7B). Phosphorylation of AMPK at threonine 172 (p-T172-AMPK) increased in MOD vs. Sham, whereas p-S485/491-AMPK increased in MOD and SHF vs. Sham. Phospholamban phosphorylation (p-S16-PLN) decreased and protein phosphatase 1a (PP1a) expression increased in MOD and SHF vs. Sham. PP1a expression up-trended further in SHF vs. MOD. Transitioning from MOD to SHF was associated with a decrease in the phosphorylation of downstream PKA targets at different cellular subcompartments. Glycogen synthase kinase-3 alpha (p-S21-GSK3α), glycogen synthase kinase-3 beta (p-S9-GSK3β), troponin I, and cardiac muscle (p-S23/24-TNNI3) were all decreased in SHF vs. MOD and Sham. p-T286-CaMKII down-trended in SHF vs. Sham.

In the next section, we will address changes in specific signaling pathways separately. Analysis of the shotgun proteomic datasets, including metabolic pathways and signaling pathways that changed in MOD and SHF vs. Sham, are presented in Appendix A. Analysis of the untargeted metabolomics datasets is presented in Appendix A. Changes in p-sites and the corresponding signaling pathways are presented in Appendix A.

### 2.4. FA Metabolism and mt FA β-Oxidation and Triacylglycerol (TAG) Metabolism

The FA receptor, Slc27a1, and the FA-binding protein 3 (Fabp3), which are involved in the FA uptake and intracellular transfer of long-chain FA and their acyl-CoA esters in the heart, respectively, were down-regulated in MOD and SHF vs. Sham (Appendix A). Moreover, the enzyme long-chain-FA-CoA ligase 1 (Acsl1), which is involved in the activation of long-chain-FA for both the synthesis of cellular lipids and degradation via β-oxidation, was down-regulated in MOD and SHF. The Fabp3 and Acsl1 were down-regulated in SHF > MOD vs. Sham (Appendix A). The rate-limiting enzyme, carnitine O-palmitoyltransferase 1, muscle isoform (Cpt1b), which is involved in the conversion of long-chain fatty-acyl-CoA to long-chain fatty-acyl-carnitine and their transport across the outer mitochondrial membrane, was decreased in SHF vs. Sham (Appendix A). CPT1B was predicted to be activated in IPA in MOD and SHF, SHF > MOD, due to the number of downstream molecules detected in the dataset. However, proteomics determined that Cpt1b was down-regulated in SHF vs. Sham (log2 fold change: −0.37, *p* = 0.002). Moreover, the enzyme malonyl-CoA decardoxylase (Mlycd), which is involved in the conversion of malonyl-CoA to acetyl-CoA and hence the activation of Cpt1b, was down-regulated in MOD and SHF, SHF > MOD, vs. Sham (Appendix A). Malonyl-CoA is a potent allosteric inhibitor of Cpt1b and hence FA metabolism, while acetyl-CoA carboxylase, beta isoform (Acacb), the enzyme converting acetyl-CoA to malonyl-CoA, was down-regulated in SHF only (Excel file 1). The specific acyl-CoA dehydrogenases, short-chain (Acadsb), medium-chain (Acadm), long-chain (Acadl), and family members 8 (Acad8) and 10 (Acad10), were downregulated in SHF vs. Sham (Appendix A). This is consistent with the metabolomic data showing all forms of acyl-carnitines to have decreased in abundance in SHF vs. Sham. PGC-1α expression and signaling decreased in SHF > MOD vs. Sham. Acetyl-CoA abundance down-trended in MOD and SHF vs. Sham (Appendix A; acetyl-CoA relative expression decreased in one Sham biological sample and statistical significance could not be attained unless that sample was excluded). Glycerol-3-phosphate dehydrogenase (Gpd1), an enzyme needed for glycerol-3-phospate generation and de novo TAG biosynthesis, was down-regulated in SHF (Appendix A). Metabolomics data showed that TAG abundance decreased in MOD and SHF vs. Sham. The metabolites TG(38:4) and TG(36:1) were down-regulated in SHF vs. Sham, whereas TG(44:2) was up-regulated, and DG(22:3) was down-regulated in MOD vs. Sham (Appendix A).

### 2.5. Glucose, Pyruvate Metabolism, and BCAA Catabolism

The glucose uptake receptor, Glut4 (Slc2a4), the predominant isoform in the adult heart than Glut1, was equally down-regulated in MOD and SHF (Appendix A). Glut4 activity is also enhanced through its phosphorylation by AMPK, which also triggers its translocation to the plasma membrane. AMPK kinase signaling was attenuated in SHF, suggesting a further decrease in glucose uptake in SHF. The metabolites 2-deoxy-D-glucose-6-phosphate and glyceraldehyde-3-phosphate were down-regulated in SHF vs. Sham and down-trended in MOD vs. Sham (Appendix A). Phosphoglucomutase (Pgm1) was up-regulated in MOD vs. Sham and was mildly increased in SHF vs. Sham. Glycogen synthase 1 (Gys1) and glycogen phosphorylase (Pygm) are two nodal enzymes that regulate glycogen homeostasis in the heart by promoting glycogen synthesis and glycogen degradation, respectively. Pygm was down-regulated in SHF vs. Sham only (Appendix A). It has been shown that Pygm is activated through its phosphorylation, either by PKA or by the calcium-activated phosphorylase kinase [20], while the serine/threonine-protein phosphatase 1 (PP1) dephosphorylates Gys1 and Pygm, promoting Gys1 activation and Pygm inhibition, respectively [21]. PP1-beta catalytic subunit (Ppp1cb) was up-regulated in SHF > MOD and PP1 expression increased in MOD and SHF. Moreover, p-S641-Gys1 was attenuated in MOD vs. Sham and p-S698-Gys1 was attenuated in SHF vs. Sham (Appendix A). These data suggest an increase in glycogen synthesis in MOD and SHF, along with enhanced glycogen degradation in MOD, and a paradoxical net increase in glycogen content in SHF > MOD. The untargeted metabolomics data showed a similar pattern of decrease in glycogen content in MOD vs. Sham and an increase in glycogen content in SHF vs. MOD (Appendix A).

The phosphofructokinase (Pfkm) enzyme, which is critical in glycolysis catalyzing the initial step in the conversion of fructose-6-phosphate to fructose-1,6-biphosphate, was down-regulated in MOD and SHF, SHF > MOD (Appendix A). Moreover, the enzymes phosphoglycerate mutase (Pgam2) and phosphopyruvate hydratase (Eno3) were down-regulated in MOD and SHF, SHF > MOD, vs. Sham (Appendix A). Additionally, the enzymes fructose biphosphate aldolase (Aldoa) and pyruvate kinase (Pkm) were downregulated in SHF vs. Sham. Pkm, which converts phosphoenoylpyruvate to pyruvate, catalyzes the second ATP-generating step in the glycolytic pathway. These data suggest that glycolysis was impaired in MOD and SHF, SHF > MOD. Pyruvate is then translocated into the mitochondria by the mt pyruvate carriers 1 (Mpc1) and 2 (Mpc2). Mpc1, pyruvate carboxylase (Pc), and the dihydrolipoamide acetyltransferase component of the pyruvate dehydrogenase (PDH) complex were down-regulated in MOD vs. Sham. In addition to the aforementioned proteins, Mpc2 and the PDH E1 component subunits alpha (Pdha2) and beta (Pdhb) were downregulated in SHF vs. Sham (Appendix A). These data suggest that transitioning from MOD to SHF was associated with a significant impairment in pyruvate metabolism.

The enzyme BCAA aminotransferase (Bcat), which catalyzes the initial step converting BCAA to branched-chain alpha-ketoacids (BCKAs), was down-regulated in MOD and SHF, SHF > MOD, vs. Sham (Appendix A). BCKAs are then irreversibly decarboxylated by the BCKA dehydrogenase (Bckdh) complex, which was mainly down-regulated in SHF. The Bckdh complex is also phosphorylated and thus inhibited by the kinase Bckdk, which was about equally down-regulated in MOD and SHF. BCKAs, through a series of reactions, are eventually catabolized into acetyl-CoA for oxidation in the TCA cycle. The enzymes responsible for these reactions were down-regulated in SHF > MOD (Appendix A). These findings were consistent with the untargeted metabolomics data suggesting that BCAAs catabolism was decreased in SHF > MOD vs. Sham.

### 2.6. AA, Ketone Body and Nucleotide Metabolism

The untargeted metabolomics data showed that the abundance of taurine and the N-acyl-amines, N-stearoyl-taurine and N-Docosanoyl-taurine, decreased in MOD vs. Sham. Tauropine and N-linoleoyl-taurine abundance equally decreased in MOD and SHF vs. Sham (Appendix A). Creatine abundance decreased in MOD and further decreased in SHF vs. MOD and Sham, whereas 3-Sulfino-alanine increased in MOD and further increased in SHF vs. MOD and Sham (Appendix A). Glutamine abundance decreased in SHF only (Appendix A). The AA, pantetheine-4-phosphate, abundance decreased in MOD and further down-trended in SHF vs. MOD (Appendix A). In addition to AA, many metabolites belonging to the glycerolipid and glycerophospholipid classes were identified and their abundance decreased in SHF vs. Sham (Appendix A), suggesting a significant alteration in cell membrane fluidity and permeability.

Metabolic pathways related to ketogenesis and ketolysis were downregulated in SHF > MOD. The enzymes acetyl-CoA acyltransferase 2 (Acaa2) and 3-hydroxybutyrate dehydrogenase 1 (Bdh1) were downregulated in MOD and SHF, SHF > MOD (Appendix A). Hdha was about equally down-regulated in MOD and SHF, whereas Hdhb was downregulated in MOD and further decreased in SHF vs. MOD and Sham. Acetoacetic acid abundance increased in SHF vs. Sham and up-trended in MOD vs. Sham (Appendix A), confirming a decrease in ketone body metabolism, as the proteomics data indicated.

The enzyme, purine nucleoside phosphorylase (Pnp), was equally up-regulated in MOD and SHF. The enzyme, 5’-nucleotidase, cytosolic II (Nt5c2), was upregulated in MOD and further increased in SHF (Appendix A). Transitioning to SHF was associated with the up-regulation of the enzymes adenosine deaminase (Ada), guanine deaminase (Gda), and xanthine dehydrogenase (Xdh) (Appendix A). The metabolite 2’,3’ cyclic AMP decreased in MOD and SHF vs. Sham and further down-trended in SHF vs. MOD (Figure 4E and Appendix A).

### 2.7. Mitochondrial Proteome and Oxidative Capacity

Succinate dehydrogenase complexes (Sdha and Sdhb) and the succinate CoA ligases (Sucla2 and Suclg1) were mildly downregulated in MOD vs. Sham and further downregulated, including Sdhc, in SHF. Additionally, all enzymes involved in the TCA cycle were down-regulated in SHF vs. Sham (Appendix A). Similarly, ETC complexes II, IV, and ATP synthase expression decreased in MOD vs. Sham. All the ETC complexes and ATP synthase were down-regulated in transitioning to SHF (Appendix A). The post-translational modification (PTM) of ATP synthase was downregulated in MOD and SHF at specific p-sites, as shown in (Appendix A). The NAD-dependent protein deacetylases (Sirt 3 and Sirt5) were down-regulated in SHF only vs. Sham (Appendix A). Moreover, the metabolite nicotinamide ribotide, also known as nicotinamide mononucleotide (NMN), which is an intermediate in NAD+ biosynthesis, down-trended in SHF vs. Sham (*p* = 0.05, Appendix A); and the enzyme nicotinamide-nucleotide adenylyltransferase (Nmnat3), which catalyzes the reaction forming NAD+ from NMN and ATP, down-regulated in SHF vs. Sham (Appendix A). All in all, these data suggest that transitioning from MOD to SHF is associated not only with down-regulation but also with PTM, including the phosphorylation and hyperacetylation of the OXPHOS machinery.

In addition to the changes related to OXPHOS, the mt contact site and cristae organizing system (MICOS), mt dynamics proteins mitofusin 1 (Mfn1) and optic atrophy 1 (Opa1), were down-regulated in SHF relative to Sham. The mt ribosomal proteins and the Slc25 mt carrier family proteins were down-regulated in SHF > MOD vs. Sham. Particularly, the ADP/ATP translocase 1 (Slc25a4) and the mt CoA transporter (Slc25a42) were equally down-regulated in MOD and SHF vs. Sham. The mt tricarboxylate transporter (Slc25a1), the mt phosphate carrier (Slc25a3), and the mt 2-oxoglutarate/malate carrier (Slc25a11) were down-regulated in SHF vs. Sham. Moreover, there was PTM of Slc25a4 with a predominant decrease in mt protein phosphorylation in SHF vs. Sham (Appendix A). Proteins involved in mt calcium uptake and efflux were predominantly affected in SHF and there was a PTM (phosphorylation) of the voltage-dependent anion channel 1 (VDAC1) rather than a change in its expression. The phosphosites, p-S57-VDAC1 and p-S241-VDAC1, were down-regulated in SHF vs. Sham (Appendix A). The mt calcium uniporter (MCU) complex and the mt proton/calcium exchanger (Letm1) were down-regulated in SHF vs. Sham (Appendix A). Phosphosites of mt proteins that changed in MOD and SHF vs. Sham are presented in Appendix A and Appendix A.

### 2.8. PKA and AMPK Signaling and their Implication in Metabolic Remodeling

PKA and AMPK are two important intracellular signaling pathways implicated in myocardial metabolism. PKA phosphorylation at Ser339 was mildly attenuated in both MOD and SHF vs. Sham by Scaffold analysis. The adenylate cyclase type 5 (Adcy5), the PKA type II regulatory subunit alpha (Prkar2a), and the A-kinase anchor protein 12 (AKAP12) were down-regulated in SHF vs. Sham, while AKAP13 was upregulated in SHF vs. Sham (Appendix A). The metabolite 2′,3′ cAMP was downregulated in MOD and further down-trended in SHF. The enzyme 2’,3’-cyclic-nucleotide 3’-phosphodiesterase (Cnp) was up-regulated in SHF vs. Sham, whereas p-S152-Pde4a and p-S45-Pde7b were down-regulated in SHF vs. Sham. The p-S16-PLN and p-S101-AKAP1 were decreased in MOD vs. Sham, while p-S103-AKAP1 down-trended in MOD and SHF vs. Sham (Appendix A). These data suggest a decrease in PKA activity at the ER–mt interface subcompartment in MOD and a more pronounced decrease in PKA signaling at multiple cellular subcompartments in SHF.

The AMPK non-catalytic subunit beta 2 (p-S107-Prkab2) was up-regulated in MOD vs. Sham. The AMPK catalytic subunit alpha 2 (p-S377-Prkaa2) was downregulated, i.e., suggested to be inhibited, in SHF relative to Sham, whereas protein phosphatase 2 regulatory subunit B’alpha (p-S41-Ppp2r5a) was up-regulated in SHF vs. Sham (Appendix A). Previous work showed that the protein phosphatase 2 regulatory subunit B delta isoform (Ppp2r2d) dephosphorylates Thr172 AMPK residue and inactivates it [22]. The p-S25-Acaca, a known downstream target of AMPK, was down-regulated in SHF vs. Sham (Appendix A), and p-S485/491-AMPK was enhanced in MOD and SHF vs. Sham, while p-T172-AMPK was enhanced in MOD vs. Sham and then decreased in SHF vs. MOD.

### 2.9. Signaling Implicated in Mitochondrial Function other than PKA and AMPK

The protein kinase c delta type, Prkcd, was upregulated in MOD and further increased in SHF vs. Sham. Additionally, protein kinase c alpha type, Prkca, was upregulated in SHF only vs. Sham (Appendix A. The p-S661-Prkcd was up-regulated and p-T521-Prkce was down-regulated in SHF vs. Sham (Appendix A). MAPK signaling was activated in MOD and SHF, SHF > MOD. The mTOR signaling was activated in SHF vs. Sham. Serine/threonine protein kinases and phosphatases that changed in MOD and SHF vs. Sham are presented in Appendix A and Appendix A.

Calcium cycling was mainly impaired in SHF. Calcium cycling at the ER–mt interface was attenuated in SHF > MOD. SERCA2a expression was mildly decreased in MOD (log2 fold change: −0.2, *p* = 0.037) and further decreased in SHF vs. Sham (Appendix A). p-S16-PLN was attenuated in both MOD and SHF vs. Sham. MCU complex proteins (Mcu and Micu1) were downregulated in SHF only (Appendix A). The Letm1 was mildly decreased in SHF vs. Sham. Calreticulin (Calr) was upregulated in SHF, while calsequestrin-2 (Casq2), Prkar1a, Prkar2a, and Camk2a were downregulated in SHF vs. Sham (Appendix A). Additionally, the plasma membrane calcium entry and efflux proteins, the voltage-dependent calcium channel (Cacna2d2) that regulates L-type calcium channel activity, and the sodium/calcium exchanger (Slc8a1) were upregulated in SHF vs. Sham, respectively (Appendix A). Moreover, p-S339-Prkaca, p-S528-Camk2d, p-S51-Vdac1, and p-S241-Vdac1 were downregulated and p-S282-Slc8a1 was upregulated in SHF vs. Sham (Appendix A). Collectively, these data suggest predominant impairment in calcium signaling and homeostasis as well as mt calcium uptake and efflux in SHF vs. Sham.

## 3. Discussion

This multi-omics-based study provides a global assessment focused on metabolic remodeling and its contribution to myocardial remodeling in a rat model of moderate remodeling and early systolic dysfunction (MOD) and in progression to SHF. It highlights the complexity of signaling pathways, their cross talk, and how metabolic intermediates regulate calcium signaling, and homeostasis, cardiac metabolism, and mt function (Figure 8). We show that the mitochondria are central in the metabolic remodeling process where metabolic pathways converge to eventually undergo OXPHOS and ATP production. We have validated that the mt proteome is predominantly down-regulated or degraded in cardiac-stress-induced myocardial dysfunction, as previously shown [23]. However, our study is also unique in that we have highlighted mt proteomic changes in a less severe MOD phenotype, as well as in an overt SHF phenotype. In both MOD and SHF, mt proteins constituted about 50% of the proteins that were down-regulated relative to Sham. Transitioning from MOD to SHF was associated with at least a 1.5-fold increase in the number and the degree of down-regulation of the commonly identified mt proteins. Moreover, other mt physiological processes, such as mt calcium uptake and efflux regulation, MICOS and the mt import system, and mt translation–elongation machinery were predominantly impaired/down-regulated in SHF. These findings support the concept that the degree of mt dysfunction develops in parallel and mirrors the degree of myocardial remodeling and progression to overt systolic dysfunction, suggesting that altered mt function is the cause rather than the consequence of HF progression, as was previously described [11]. We highlight molecular mechanisms leading to progression in mt dysfunction in HF. Our findings suggest that mt pathophysiological changes in MOD are predominantly related to attenuation in mt FA β-oxidation, BCAA catabolism and OXPHOS. Upon progression to SHF, there is a worsening of the aforementioned pathophysiological processes. Additionally, there is impaired pyruvate metabolism, enhanced mt degradation and apoptosis (impaired mt calcium regulation and reactive oxygen species (ROS) production) [5,8,9,24,25], and derangements in the mt import systems. These mt pathophysiological changes constitute the drive for the worsening in mt function and myocardial remodeling, and the transition to an advanced stage of HF.

About two-thirds of the proteins involved in metabolism and respiration that were down-regulated in MOD and SHF vs. Sham were mt proteins. Impairments in OXPHOS and mt FA β-oxidation were conspicuous in SHF > MOD, as previously shown [6]. Furthermore, our data suggest that OXPHOS and mt FA β-oxidation were the most enriched metabolic pathways that were inactivated in SHF > MOD. We showed that PTM (phosphorylation and hyperacetylation) of the OXPHOS machinery and mt proteins was more predominantly evident in SHF than MOD. Our findings are consistent with previous acetylproteomics work showing extensive mt protein lysine hyperacetylation in mouse models of early-stage HF and in end-stage human HF [26,27]. Moreover, BCAA catabolism, AA metabolism (tryptophan and glutamine degradation), and the Krebs cycle were highly enriched metabolic pathways that were attenuated in SHF > MOD. Previous work showed that the accumulation of BCAAs exerts detrimental effects on the myocardium by enhancing mTOR signaling (therefore suppressing autophagy) and ROS production in the mitochondria [28,29]. Moreover, BCAA accumulation exerts an inhibitory effect on the PDH complex by inhibiting the hexosamine biosynthetic pathway, which suppresses pyruvate metabolism and sensitizes the heart to cardiac stress [30].

Glucose metabolism was differentially regulated in MOD and SHF. Glycogen and glucose-1-phosphate degradation were activated in MOD > SHF, while glucose uptake was attenuated in SHF > MOD. Glut 4 expression was equally down-regulated in MOD and SHF; however, its AMPK-mediated PTM (activation) was attenuated in SHF only. Moreover, glycolysis and pyruvate metabolism were predominantly inactivated in SHF rather than MOD. Overall, our findings indicate that glucose metabolism is mainly impaired in SHF, as previously shown [6], but with some caveats that are noteworthy to highlight. While glucose uptake was attenuated in MOD, it seems that the heart compensates at the earliest stages of systolic dysfunction by up-regulating pathways involved in endogenous glycogen degradation and glucose biosynthesis, which are then attenuated in SHF along with significant decreases in glycolysis and pyruvate metabolism. This leads to a paradoxical net increase in glycogen content. The metabolomics data suggest a pattern of decrease in glycogen content in MOD vs. Sham and an increase in glycogen content in SHF vs. MOD. Glycogen is an important energy source in the heart, which is quickly mobilized when needed, especially under acute changes in working load. Glycogen occupies about 2% of the adult and 30% of the fetal cardiomyocyte volume [31]. Unlike liver and skeletal muscle, the heart increases its glycogen content during fasting because free FA (FFA) become the predominant substrates for oxidation and inhibit glycolysis, thus routing glucose for glycogen synthesis [32]. An increase in glycogen content in advanced HF may play a detrimental role, as has been shown in high-glycogen-content ischemic rat hearts [33], and in the extreme cases of glycogen storage diseases [34]. Based on our findings, we believe that this is likely related to the dysregulation of pathways governing glycogen homeostasis and glucose metabolism, rather than just an increase in glycogen content per se.

Our data show unique findings in terms of TAG homeostasis and the accumulation of cardiolipotoxic material. Despite that mt FA β-oxidation and TAG degradation were attenuated in SHF > MOD, the abundance of ceramides and TAG-related metabolites was attenuated in both MOD and SHF. Previous work has shown that FFA uptake into the cardiac myocyte is increased in the advanced stage of HF due to enhanced sympathetic activation and an increase in plasma FFA levels [35]. Mismatch between enhanced FFA uptake and their oxidation leads to the accumulation of toxic lipid species, such as ceramides and diacylglycerol (DAG), and lipotoxicity-induced mt dysfunction and apoptosis in advanced HF [36,37]. Our findings in MOD and SHF do not support this hypothesis, but rather point toward a decrease in FA uptake and FA processing in MOD and SHF, SHF > MOD. The FA uptake receptor (Slc27a1) was equally down-regulated in MOD and SHF, while the protein, Fabp3, and the enzyme, Acsl1, involved in FA transport and activation, respectively, were down-regulated in SHF > MOD. Our findings are consistent with previous work showing a decrease in endogenous TAG content, turnover, and oxidation in rat models of PO-induced early cardiac failure [38], and volume overload-induced HF and eccentric remodeling [39].

Our data highlight important metabolites and metabolite intermediates that play a role in myocardial function and metabolism rather than in ATP production. Taurine and creatine constitute two important AAs that play a role in cardiomyocyte function, signaling, and metabolism. Taurine and its derivatives’ abundance decreased in MOD > SHF, while creatine abundance decreased in SHF > MOD. Taurine deficiency in mice, through the genetic knockout of the taurine transporter (TauT) and TauT inhibition in rats was associated with cardiomyopathy [40]. Taurine deficiency was associated with reduced glucose and FA oxidation in isolated perfused rat heart [41], and reduced ETC complexes I and III activity with increased ROS production [42], as well as altered calcium homeostasis and signaling [43]. Taurine supplementation was shown to be beneficial in HF, and was associated with improvement in LV function [44] and exercise capacity [45]. Pantetheine-4-phosphate abundance decreased in MOD and down-trended in SHF vs. MOD. Pantetheine, also known as pantothenate, is phosphorylated by the pantothenate kinase to pantetheine-4-phosphate, which is the first step in the biosynthesis of CoA and is also used to shuttle intermediates between the active sites of enzymes involved in FA peptide and polyketide synthesis, thus playing role in cell growth and metabolism [46]. Eicosanoids, which are metabolites of arachidonic acid, were down-regulated in SHF vs. Sham. These metabolites have been shown to play an important role in calcium homeostasis and cycling and cardiac contraction through the regulation/inhibition of the L-type calcium channels (LTCC) [47] and the calcium-independent phosphorylation of troponin I and myosin light chain 2 [48].

Our data suggest an enhanced nucleotide degradation, including 2′,3′ cAMP (or 3′,5′ cAMP), in SHF > MOD. Currently, we do not know whether an increase in nucleotide metabolism is an adaptive versus maladaptive process in HF. However, previous work and our data suggest that enhanced nucleotide metabolism play a detrimental rather than beneficial role and contribute to myocardial dysfunction and metabolic remodeling. Enhanced Xdh activity (Xdh expression increased in SHF) was shown to be associated with enhanced ROS and that its inhibition may be beneficial in the context of HF [49,50]. Moreover, our data show that ketone body oxidation was attenuated in SHF, contrary to what was previously shown in animal models [51] and in human end-stage HF [52].

PKA phosphorylation at Ser339 residue was mildly attenuated in both MOD and SHF vs. Sham by Scaffold analysis. Previous work has shown that PKA phosphorylation at Ser338 (reported as Ser339 in PosphoSitePlus.org) precedes PTM of the activation loop Thr197, and is required for PKA processing and maturation as well as for its phosphorylation at Thr197 [53]. In MOD, PKA signaling was more ambiguous, showing enhanced phosphorylation of some PKA targets: ryanodine receptor isoform 2 (p-T1850-Ryr2, not previously reported) and myosin light chain 3 (p-S111-Myl3, not previously reported), as well as decreased phosphorylation of other targets: p-S641-Gys1, and p-S14-Myl2 (Appendix A). These data suggest a differential change in PKA activity at specific cellular subcompartments in MOD (as highlighted in the subsequent section), rather than a global intracellular decline in PKA activity, and that the ER–mt interface may constitute one of the important compartments where PKA activity is down-regulated at the initial stage of cardiac remodeling and systolic dysfunction.

P-AMPK-T172 was increased in MOD, while p-S485/491-AMPK was increased in MOD and SHF vs. Sham. Previous work has shown that AMPK activation is mediated through its phosphorylation at Thr172 [54,55,56], while its phosphorylation at Ser485/491 was identified as being an auto-phosphorylation site, and a target site for protein kinase B/AKT, which is responsible for the insulin-mediated inhibition of AMPK upon insulin stimulation [57,58]. PKA has also been shown to phosphorylate AMPK at Ser485/491 and to limit its activation in response to energy depletion or other regulators [56]. The collective data suggest that AMPK activity may be enhanced or unaltered in MOD and is attenuated in SHF.

Cross talk between PKA and AMPK signaling pathways exists. Both regulate common pathways involved in carbohydrate, lipid, and protein metabolism [17,59,60,61]. Our data show derangements in FA, glucose metabolism, and OXPHOS, and a decline in PGC-1α and PPARα signaling that was evident in MOD, despite the fact that AMPK signaling was not affected in MOD. This is likely explained by derangements in PKA signaling in specific cellular subcompartments in MOD due to heterogeneity in the localization or activity of the adenylate cyclases, phosphodiesterases, AKAPs, and phosphatases, such as PP1a-induced dephosphorylation of common PKA targets in specific cellular subcompartments as previously shown [16,62,63,64]. We show that PP1a expression was increased in MOD and SHF vs. Sham and up-trended further in SHF vs. MOD. The p-S101-AKAP1, a scaffold protein localizing PKA to the outer mt membrane (OMM), was attenuated in MOD vs. Sham, while p-S103-AKAP1 down-trended in both MOD and SHF vs. Sham. Previous work has shown that AMPK phosphorylates AKAP1 at Ser103 [65], while PP1 destabilizes AKAP1, promoting its degradation [66]. These data suggest a decrease in mt PKA activity at the OMM in MOD and SHF, SHF > MOD. Mitochondrially targeted PKA have been shown to be beneficial for mt function by: 1) promoting p-S-637-DRP1, i.e., inhibiting mt fission [67,68]; 2) enhancing the activity of the mt import system [69,70,71]; and 3) by directly enhancing the activity of the ETC complexes I and IV, and ATP synthase, and therefore OXPHOS and ATP production [72,73,74,75].

Besides PKA, protein kinase C isoforms: alpha (Prkca), delta (Prkcd), and epsilon (Prkce) are known to localize to mitochondria and to regulate mt function, metabolism and apoptosis [76,77]. PKC isoform activity is dependent upon their expression level, localization, and phosphorylation status [78]. Our data suggest enhanced Prkcd signaling in SHF > MOD and a decrease in Prkce signaling in SHF. Prkcd is known to contribute to stress-induced mt dysfunction in neurons [76] and is a critical mediator of post-ischemic cardiomyocyte necrosis and contractile dysfunction [79]. Moreover, Prkcd has been shown to exert a pro-hypertrophic role in the non-stressed heart [79]. Prkce plays more of a protective role in the heart and neurons [76]. Increased Prkce activity or expression in the heart has been shown to promote adaptive hypertrophy and eutrophic cardiac growth [79]. Additionally, mt-targeted Prkce has been shown to phosphorylate ETC complex IV and to increase its activity in the heart and kidney in response to stress [78,80]. Similar to Prkcd, Prkca has been shown to play a role in maladaptive hypertrophy and cardiac dysfunction [79]. MAPKs can also translocate to the OMM and modulate mt function. The MAPK, c-Jun N-terminal kinase (JNK), which is activated in cardiac stress [10], is of particular interest as it has been shown to localize to the mitochondria via its interaction with the scaffold protein Sab [81] and to phosphorylate mitofusin 2 at Ser27 (promoting its degradation and mt fission) [82], and the bcl-2 family proteins, thereby promoting mt apoptosis [10,76,83].

In conclusion, our data provide strong evidence that metabolic remodeling plays a causative role in myocardial remodeling and progression to overt SHF. We highlight changes in the mt proteome and p-proteome in early systolic dysfunction and in overt SHF. Although decline in mt OXPHOS and mt FA β-oxidation and BCAA catabolism constitute the initial mt pathophysiological processes that are impaired in early systolic dysfunction, progression to SHF is accompanied by fundamental impairment in mt function at multiple regulatory levels, including calcium and redox balance regulation, derangements in MICOS, mt import, and mt translation–elongation systems. We have highlighted the role of important AAs and metabolites in the regulation of myocardial signaling and calcium cycling and homeostasis in both MOD and SHF. Finally, we have highlighted important myocardial signaling pathways and the complexities that govern metabolic remodeling and derangements in calcium cycling in progression to SHF. Schematic drawings of important findings in MOD and SHF vs. Sham are presented in Figure 8A–C and Appendix A. Our findings are important because they provide not only a global understanding of important myocardial pathophysiological processes involved in metabolic remodeling, but also a framework for future comprehensive mechanistic studies for exploring and identifying potential therapeutic targets in HF. Strategies restoring ER–mt–PKA signaling may constitute a potential therapeutic strategy for HF that is worth investigating in the future.

## 4. Materials and Methods

### 4.1. Experimental Model of Ascending Aortic Banding for the Creation of the MOD and SHF Phenotypes

All procedures involving the handling of animals were approved by the Animal Care and Use Committee of Tulane University and adhered to the National Institutes of Health Guide for the Care and Use of Laboratory Animals (Protocol 812, approval date 25 September 2019). The MOD and SHF phenotypes were created using the AAB procedure in Sprague Dawley rats, as previously described [24,84]. Animals underwent transthoracic echocardiography as previously described [84], to assess LV size and function and assure the presence of compensated concentric hypertrophy at week 3 post-AAB, and the development of MOD and SHF phenotypes at week 8 post-AAB. The creation and characterization, by echocardiography and hemodynamics, of these two models of HF have been described elsewhere [85]. Left ventricular myocardial tissue from the same biological samples were used to perform the proteomics and metabolomics studies; *n* = 3 biological samples per group. Additionally, the same Sham biological samples were used for both MOD and SHF proteomics and p-proteomics runs.

### 4.2. Discovery-Based Proteomic and p-Proteomic Analysis Using Tandem Mass Tags and Liquid Chromatography–Mass Spectrometry

Description of the methods for protein extraction, in-solution proteolytic digestion and tandem mass tags (TMT)-labeling, global TMT sample fractionation and Orbitrap Fusion liquid chromatography–mass spectrometry (LC–MS) peptide analysis and database searching are presented in the online Appendix A.

### 4.3. Criteria for Protein Odentification

Scaffold (version 4.9, Proteome Software Inc., Portland, OR, USA) was used to validate tandem MS-based peptide and protein identifications. Peptide and protein false discovery rates (FDRs) were set to 1% and the minimum number of identified peptides was set to 2. Protein probabilities were calculated by the Protein Prophet algorithm [86]. Proteins that contained similar peptides and could not be differentiated based on MS/MS analysis alone were grouped to satisfy the principle of parsimony. Proteins sharing significant peptide evidence were grouped into clusters.

### 4.4. Protein Quantification in Scaffold

Scaffold Q+ (version 4.9, Proteome Software Inc., Portland, OR, USA) was used for TMT-based peptide and protein quantification. Channels were corrected according to the algorithm described in i-Tracker [87] in all samples. Normalization was then performed on intensities across samples and spectra, as described in the statistical analysis of relative labeled mass spectrometry data from complex samples using ANOVA [88]. Median values were used for averaging. Spectra data were then log2-transformed, pruned of those matched to multiple proteins, and weighted by an adaptive intensity-weighting algorithm. Of the 64,264 spectra in the experiment at the given thresholds, 36,096 (56%) were included in quantitation. Differentially expressed proteins across experimental groups were determined by applying a permutation test with Benjamini–Hochberg correction for multiple comparisons [89]. An unadjusted *p* value of < 0.05 was considered significant.

MS raw, log2-transformed data were exported from Scaffold and uploaded into Qlucore bioinformatics software for data visualization and presentation of the dataset in Scaffold via the generation of heat maps, PCA plots, and Venn diagrams. Additionally, the uploaded MS raw, log2-transformed datasets were statistically analyzed in Qlucore bioinformatics software by two-group comparison T-test; *p* < 0.05 was considered statistically significant. Heat maps, volcano plots, PCA plots, and Venn diagrams were then generated for data visualization and presentation of the statistically analyzed dataset in Qlucore.

### 4.5. Bioinformatic Analyses

Statistically analyzed datasets in Scaffold and Qlucore bioinformatic software programs, including all identified proteins in the proteomic TMT multiplex experiments, were uploaded into IPA bioinformatics software. A core analysis was performed for each of the two-group comparisons (MOD vs. Sham and SHF vs. Sham) using the Ingenuity Knowledge database and a cutoff *p*-value of < 0.05. Analyzed datasets were then compared using IPA “Comparison Analyses” function to yield the most enriched Canonical Pathways, filtered to those related to metabolism and signaling as well as upstream regulators that were shared between the two-group comparisons. The generated heat maps were presented based on a cutoff -log10 *p*-value of 1.3 and a z-score of ± 2. IPA analysis content information for the MOD and SHF proteomic datasets analyzed in Qlucore were the following: Analysis Creation Date: 2021-05-03, Content version: 62,089,861 (Release Date: 17 February 2021), Analysis IDs: 33,670,484 and 33,670,824, respectively. IPA analysis content information for the MOD and SHF proteomic datasets analyzed in Scaffold were the following: Analysis Creation Date: 6 February 2021, Content version: 60,467,501 (Release Date: 19 November 2020), Analysis IDs: 27,199,584 and 27,199,664, respectively.

Description of the methods for phosphorylated Peptide TMT Enrichment, Orbitrap Fusion LC-MS phosphopeptide analysis, and database searching are presented in the online Appendix A.

### 4.6. Criteria for Protein Identification

Scaffold (version 4.9, Proteome Software Inc., Portland, OR, USA) was used to validate tandem MS-based peptide and protein identifications with the ‘mudpit’ option selected for the goal of combining the results from TiO2 and FeNTA enrichments into a single report. Peptide and protein FDRs were set to 1%, and the minimum number of identified peptides was set to 2. Protein probabilities were calculated by the Protein Prophet algorithm [86]. Proteins containing similar peptides that could not be differentiated based on an MS/MS analysis were grouped to satisfy the principle of parsimony. Proteins sharing significant peptide evidence were grouped into clusters.

### 4.7. Protein Quantification in Scaffold

Scaffold Q+ (version 4.9, Proteome Software Inc., Portland, OR, USA) was used for TMT-based peptide and protein quantification. Channels were corrected according to the algorithm described in i-Tracker [87] by correction factors in all samples. Normalization was then performed on intensities across samples and spectra, as described in the statistical analysis of relative labeled mass spectrometry data from complex samples using ANOVA [88]. Median values were used for averaging. Spectra data were log2-transformed, pruned of those matched to multiple proteins, and weighted by an adaptive intensity-weighting algorithm. Of the 43,180 spectra in the experiment at the given thresholds, 26,377 (61%) were included in quantitation. Differentially expressed proteins across experimental groups were determined by applying a permutation test with Benjamini–Hochberg correction for multiple comparisons [89]. An unadjusted *p* value of < 0.05 was considered significant.

### 4.8. Phosphopeptide Quantification in Scaffold PTM

Scaffold PTM (version 3.2, Proteome Software Inc., Portland, OR, USA) was used for TMT-based phosphopeptide quantification with the output from Scaffold Q+. Phosphopeptide results were normalized with the global protein quantification results from the related TMT experiment using proteinXML exports from Scaffold Q+. The PTM sites derived from Scaffold were annotated in Scaffold PTM using the site localization algorithm developed by Beausoleil et al. [90]. Scaffold PTM re-analyzes MS/MS spectra identified as modified peptides and calculates Ascore values and site localization probabilities to assess the level of confidence in each PTM localization. Scaffold PTM then combines localization probabilities for all peptides containing each identified PTM site to obtain the best-estimated probability that a PTM is present at that site. For motif analysis, PTMs were scanned for over-represented patterns in the amino acids surrounding the modification sites using the method described previously [91]. The background percentage was calculated using the target human protein sequence database described earlier.

Normalized raw phosphopeptide results with the global protein quantification were exported from Scaffold PTM as log2-transformed values and uploaded into Qlucore bioinformatics software for data visualization and presentation of the statistically analyzed dataset in Scaffold PTM via heat maps, PCA plots, and Venn diagrams. Additionally, the uploaded raw log2-transformed datasets were analyzed in Qlucore bioinformatics software using the two-group comparison T-test and a cutoff of *p* < 0.05 for statistical significance. Heat maps, volcano plots, PCA plots, and Venn diagrams were generated for data visualization and presentation.

### 4.9. Bioinformatic Analyses

Bioinformatic analysis was performed in IPA as described above, except that a “Phosphorylation Core Analysis” was performed by expression log ratio and a cutoff *p*-value < 0.05. Analyzed datasets were compared using IPA’s “Comparison Analyses” function to yield the most enriched signaling pathways that were present in the two-group comparisons. The generated heat maps were presented based on *p*-value and z-score. IPA analysis content information for the MOD and SHF p-proteomic datasets analyzed in Qlucore were the following: Analysis Creation Date: 2021-04-23, Content version: 62,089,861 (Release Date: 17 February 2021), Analysis IDs: 32,911,164 and 32,911,624, respectively. IPA analysis content information for the MOD and SHF p-proteomic datasets analyzed in Scaffold were the following: Analysis Creation Date: 17 April 2021 and 16 April 2021, respectively, Content version: 62,089,861 (Release Date: 17 February 2021), Analysis IDs: 32,434,384 and 32,433,404, respectively.

### 4.10. Validation by Immunoblotting

Description of the method for this section is presented in the Appendix A.

### 4.11. Discovery-Based Non-Targeted Central Carbon Metabolism and Acyl-Carnitine Analysis by LC–MS

Description of the methods for sample preparation, and central carbon and acyl-carnitine LC–MS analysis are presented in the Appendix A.

#### 4.11.1. Central Carbon Metabolism Data Identification and Normalization

Metabolites were identified by matching the retention time and mass (+/− 10 ppm) to authentic standards. Peak areas were integrated using Profinder v8.00 (Agilent Technologies, Santa Clara, CA, USA). Data were normalized to urine creatinine levels and Loess drift correction was applied using the area of each metabolite in the control samples (pools) for correction, as previously described using MetaboDrift 1.0 [92]. Non-targeted data analysis was performed using Agilent’s MassHunter Find by Molecular Feature workflow (v7.0) with recursion using Agilent’s Mass Profiler Pro (v8.0.) The dataset was processed using Binner [93] to remove degenerate features, and the resulting features were searched against the Metabolomics Workbench Refmet database (https://www.metabolomicsworkbench.org/databases/refmet/index.php), last accessed date 9 November 2021, to provide Metabolomics Standards Initiative (MSI) [94] Level III identifications, or to an in-house library of authentic standards to provide MSI Level I identifications. Duplicate identified metabolites that had the same mass, retention time, and similar abundance among the studied biological samples were removed. For example, both 2′,3′ cyclic AMP (cAMP) and 3′,5′ cAMP were identified that had the same mass, retention time, and similar abundance among the studied biological samples. These were thought to be duplicate metabolites. 2′,3′ cAMP was kept and 3′,5′ cAMP was removed.

#### 4.11.2. Acyl Carnitine Data Identification

Metabolites were identified by matching the retention time and mass (+/− 10 ppm) to authentic standards. Isotope peak areas were integrated using MassHunter Quantitative Analysis vB.07.00 (Agilent Technologies, Santa Clara, CA, USA) of metabolite fluxes for central carbon, acyl-carnitine and FA metabolites.

#### 4.11.3. Statistical Analysis of the Metabolomics MS Raw Data

MS raw data were uploaded into Qlucore bioinformatics software, and then were log2-transformed. Statistical analysis was performed in Qlucore bioinformatics software using the multi-group comparison one way ANOVA and correction by Tukey. A *p*-value < 0.05 was considered statistically significant. Then, heat maps, volcano plots, PCA plots, and Venn diagrams were generated for data visualization and presentation of the statistically analyzed dataset in Qlucore.

#### 4.11.4. Bioinformatic Analyses

Bioinformatic analysis was performed using IPA as described above, except that a “Metabolomics Core Analysis” was performed using expression log ratio and a cutoff *p*-value < 0.05. Analyzed datasets were compared using IPA’s “Comparison Analyses” function to yield the most enriched metabolic pathways and upstream regulators that were present in the two-group comparisons. The generated heat maps were presented based on *p*-value for metabolic pathways and z-score for upstream regulators. IPA analysis content information for the metabolomics dataset analyzed in Qlucore were the following: Analysis Creation Date: 23 April 2021, Content version: 62,089,861 (Release Date: 17 February 2021), Analysis ID: 32,906,044.

### 4.12. Statistical Analysis

Statistical methods used for the analysis of the proteomics, p-proteomics and metabolomics datasets were described in detail in previous sections. A *p*-value < 0.05 was considered significant. Western blot data are presented as mean ± standard deviation. Statistical analyses were performed in Prism software version 9.1.0 using one-way ANOVA with Benjamini correction method. A *p*-value < 0.05 was considered significant.

## Figures and Tables

**Figure 1 ijms-23-00235-f001:**
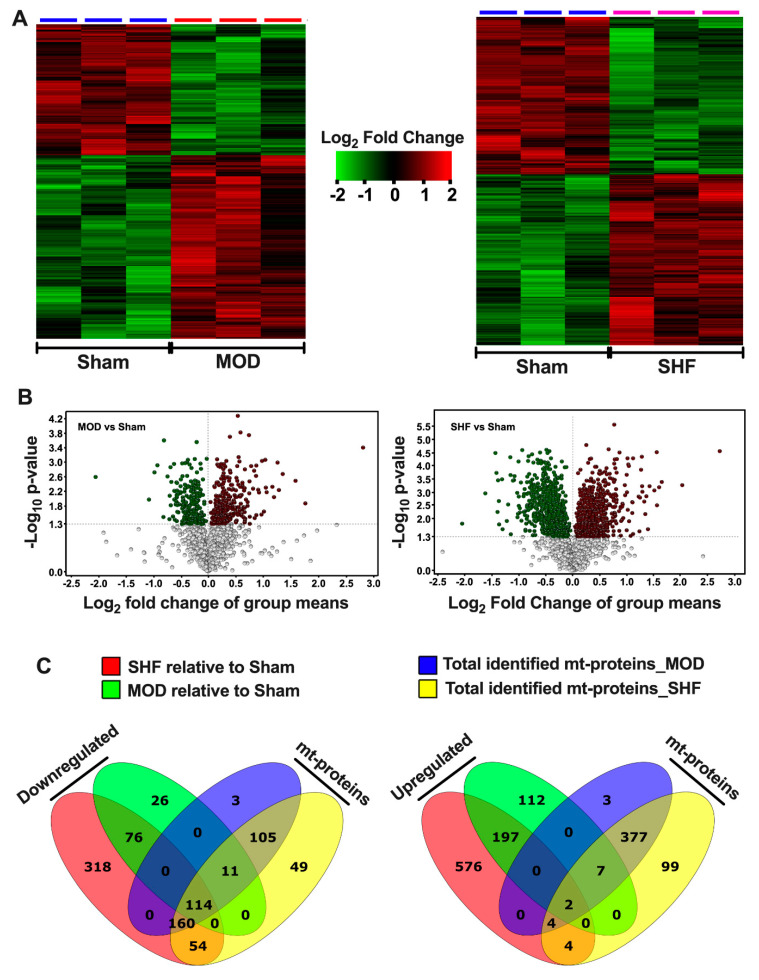
Visualization of the MOD and SHF proteomic datasets. (**A**) Heat maps showing differential log_2_ fold change in expression of identified proteins that changed in MOD vs. Sham (left), and in SHF vs. Sham (right). The same Sham biological samples were used as a reference for both TMT-label proteomic runs; *n* = 3 biological samples per group. (**B**) Volcano plots of the identified proteins that changed in MOD (left) and SHF (right) groups vs. Sham. The *x*-axis displays the log_2_ fold change in group means for down-regulated proteins (green) and up-regulated proteins (red) in MOD or SHF vs. Sham. The *y*-axis displays the -Log_10_
*p*-value of identified proteins that were statistically significant in MOD (left) and SHF (right) vs. Sham. A -Log_10_
*p*-value cutoff of 1.3 (*p* = 0.05) and higher was considered significant. (**C**) Venn diagrams showing the number of identified proteins that changed in SHF (red) and MOD (green) groups, divided into those whose relative expression decreased (left) or increased (right) vs. Sham. The blue and yellow diagrams show the total number of identified mitochondrial (mt) proteins in the MOD and SHF proteomic runs, respectively. (**D**) Heat maps showing differential log_2_ fold change in expression of proteins related to metabolism and respiration that changed in MOD vs. Sham (left), and in SHF vs. Sham (right). (**E**) Volcano plots showing the log_2_ fold change in group means (*x*-axis) and the corresponding -Log_10_
*p*-value on the *y*-axis of the identified proteins presented in (**D**). (**F**) Venn diagrams showing the total number of identified proteins related to metabolism and respiration that decreased (left) or increased (right) in MOD and SHF groups, green and red diagrams, respectively. They also show how many of those were mt proteins (intersection of green and blue diagrams for MOD and red and yellow diagrams for SHF) vs. non-mitochondrial proteins (numbers that fall outside the intersection of green and blue diagrams for MOD and red and yellow diagrams for SHF).

**Figure 2 ijms-23-00235-f002:**
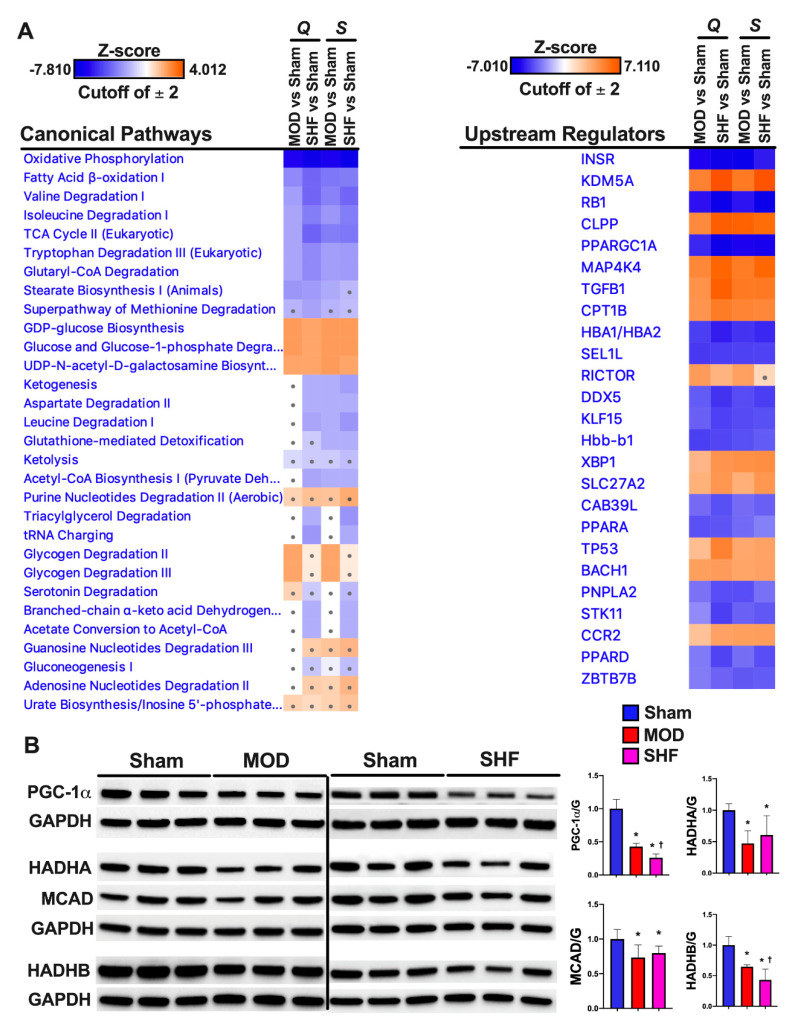
Top-ranked Canonical Pathways related to metabolism and *Upstream Regulators*, resulting from IPA’s ‘Comparison Analyses’ of the experimental groups. (**A**) ‘Core Analyses’ of the statistically analyzed proteomic datasets, by both Qlucore (Q) and Scaffold (S) bioinformatics software, were performed in IPA for each of the two-group comparisons, i.e., MOD vs. Sham and SHF vs. Sham, with a cutoff *p*-value < 0.05. Analyzed datasets were then subsequently compared with each other in IPA’s ‘Comparison Analyses’ function to yield the most enriched *Canonical Pathways* (left) related to metabolism and *Upstream Regulators* (right) that were shared among the two-group comparisons. The respective z-score-based heat maps indicate shared *Canonical Pathways* and *Upstream Regulators* that were up-regulated/activated or down-regulated/inhibited in the two-group comparisons, with orange and blue color intensities representing the z-score-based extent of up-regulation/activation or down-regulation/inhibition, respectively. *Canonical Pathways* and *Upstream Regulators* with a z-score outside the cutoff value are represented by a dot on the heat map. (**B**) Western blot in left ventricular myocardium showing the expression of the peroxisome proliferator-activated receptor gamma coactivator 1-alpha (PGC-1a), and proteins involved in mt fatty acid β-oxidation (HADHA, MCAD and HADHB), * *p* < 0.05 vs. Sham and ^†^
*p* < 0.05 vs. MOD.

**Figure 3 ijms-23-00235-f003:**
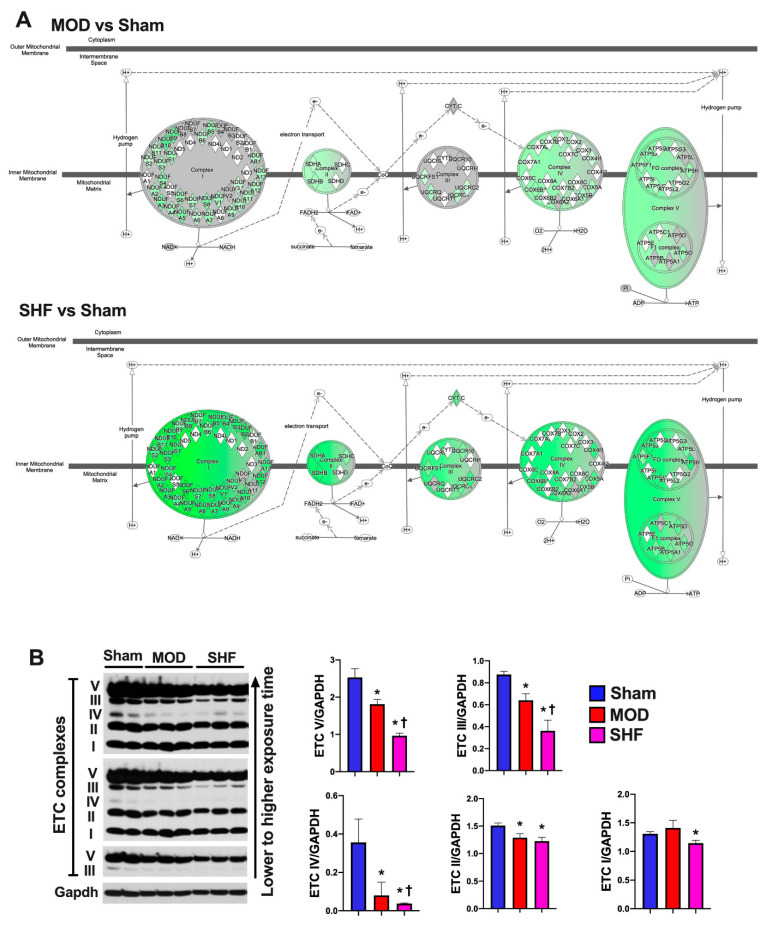
Oxidative phosphorylation in MOD and SHF. (**A**) Oxidative phosphorylation (OXPHOS) network of the electron transport chain (ETC) complexes I-IV transport system and ATP synthase are presented in MOD vs. Sham (upper panel) and SHF vs. Sham (lower panel). The network highlights mt OXPHOS proteins in each ETC complex that were downregulated (green) vs. Sham, were unchanged (grey) vs. Sham, or were not identified (white) in the MOD and SHF proteomic runs. The darker the green color, the higher the degree of downregulation. (**B**) Expression of the ETC complexes I-IV and ATP synthase (complex V) in MOD and SHF by immunoblotting, * *p* < 0.05 vs. Sham and ^†^
*p* < 0.05 vs. MOD.

**Figure 4 ijms-23-00235-f004:**
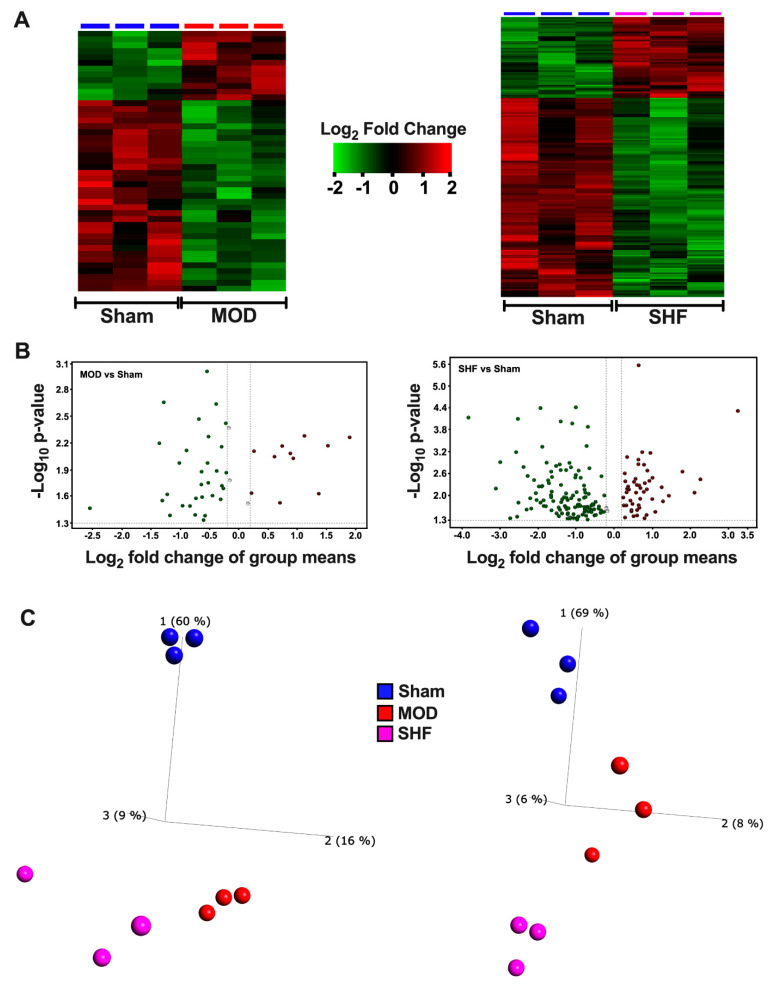
Visualization of the non-targeted metabolomics dataset in Sham, MOD, and SHF groups. (**A**) Heat maps showing differential change in log_2_ fold expression of metabolites that changed in MOD vs. Sham (left), and in SHF vs. Sham (right); *n* = 3 samples per group. (**B**) Volcano plots showing the log_2_ fold change in group means, on the *x*-axis, and the corresponding -Log_10_ (*p*-value), on the *y*-axis, for the metabolites that were downregulated (green) or upregulated (red) in MOD (left) and SHF (right) vs. Sham. A -Log_10_ (*p*-value) cutoff of 1.3 and higher was considered significant. (**C**) PCA plots showing the variance in biological samples within and between the Sham, MOD, and SHF groups for the metabolites that changed in MOD (left) and SHF (right) vs. Sham. (**D**) Venn diagram showing the number of metabolites that changed in SHF (red) and MOD (blue) vs. Sham and those that changed in SHF vs. MOD (green). (**E**) Volcano plot of the 11 common identified metabolites that changed in MOD and SHF vs. Sham (intersection of red and blue diagrams excluding the green diagram). Of those, there was a trend for further downregulation of pantetheine 4′-phosphate and 2′,3′ cyclic AMP in SHF compared with MOD. (**F**) Heat map and PCA plot for the 17 common metabolites that changed in MOD and SHF vs. Sham. (**G**) Heat map and PCA plot for the metabolites that changed in SHF vs. Sham and MOD groups. Heat maps and PCA plots in both F and G show the differential log_2_ fold change in expression in Sham, MOD, and SHF groups and the variance in biological samples, respectively, for the metabolites presented in (**F**) and (**G**).

**Figure 5 ijms-23-00235-f005:**
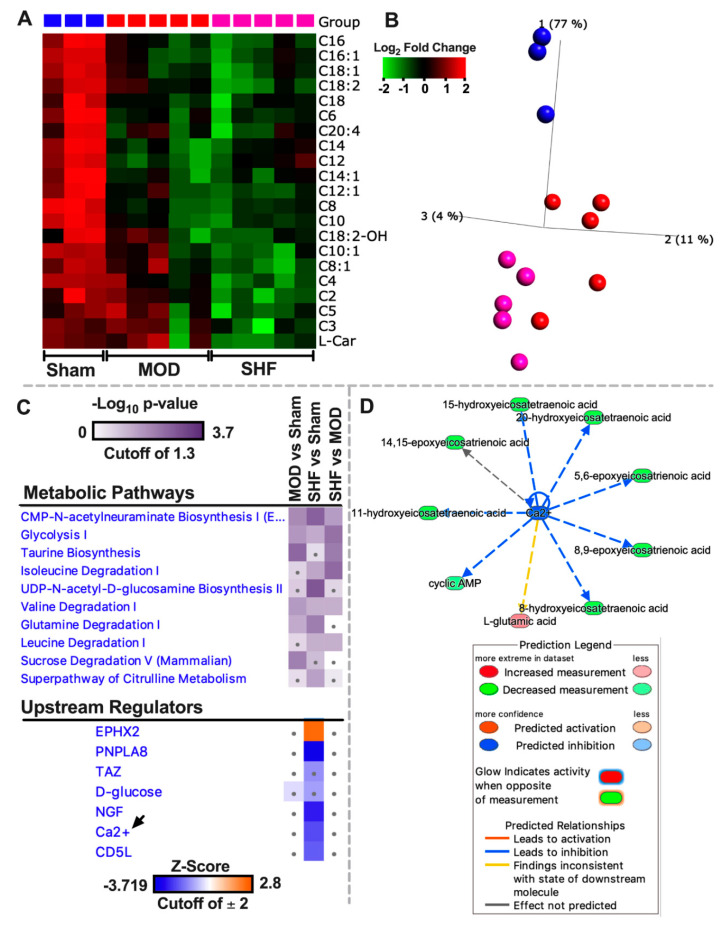
Fatty-acyl-carnitine abundance, top-ranked *Metabolic Pathways*, and *Upstream Regulators* in the MOD and SHF. (**A**,**B**) Heat map and PCA plot showing the differential log_2_ fold change in expression and variance of biological samples for fatty-acyl-carnitines that changed in MOD and SHF groups vs. Sham. (**C**) ‘Core Analyses’ of the statistically analyzed dataset, by Qlucore bioinformatics software, was performed in IPA for each of the two-group comparisons and then were compared in IPA’s ‘Comparison Analyses’ function. The generated heat maps show the most enriched *Metabolic Pathways* (upper panel) by *p*-value and *Upstream Regulators* (lower panel) by z-score. (**D**) Mechanistic network showing connection between calcium, as a second messenger, and metabolites implicated in calcium cycling and homeostasis. Please refer to prediction legend for details.

**Figure 6 ijms-23-00235-f006:**
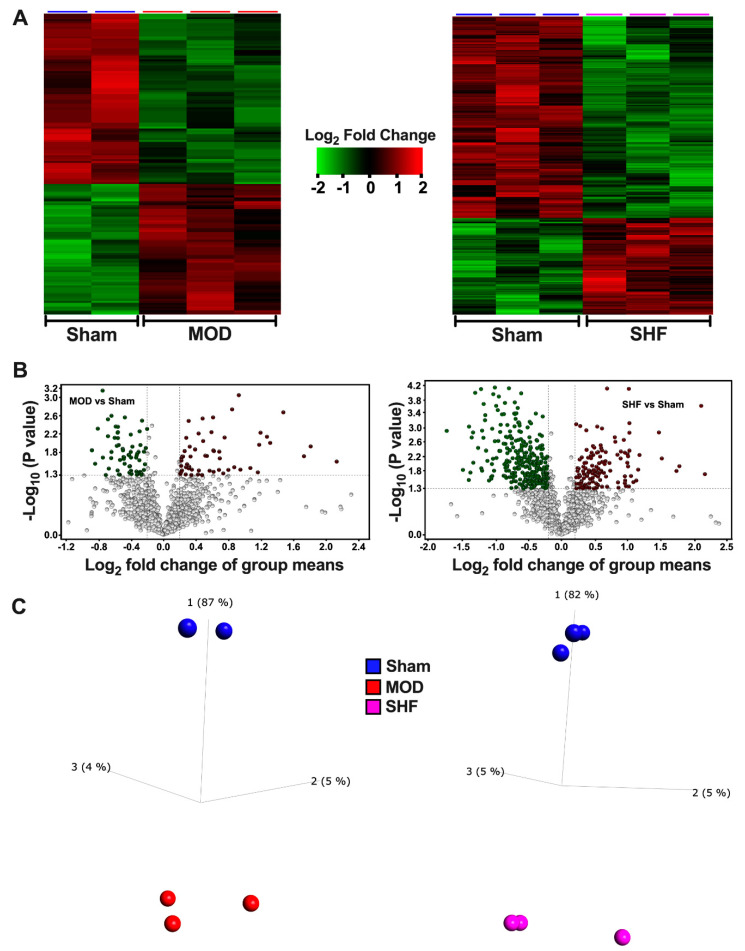
Visualization of the MOD and SHF phospho (p)-proteomic datasets. (**A**) Heat maps showing differential log_2_ fold change in the expression of p-sites that changed in MOD vs. Sham (left), and in SHF vs. Sham (right). The same biological samples in the Sham group were used as a reference for both TMT-label p-proteomic runs; *n* = 3 biological samples per group. Note: there was a technical problem in one of the Sham biological samples in the MOD p-proteomic run, which was eliminated as it was skewing the data. (**B**) Volcano plots showing the degree of log_2_ fold change, on the *x*-axis, with the corresponding -Log_10_
*p*-value, on the *y*-axis, for the p-sites that were downregulated (green) or upregulated (red) in MOD (left) and SHF (right) groups compared with Sham. (**C**) PCA plots showing the variance of biological samples in the Sham, MOD and SHF groups for the p-sites that changed in MOD (left) and SHF (right) vs. Sham.

**Figure 7 ijms-23-00235-f007:**
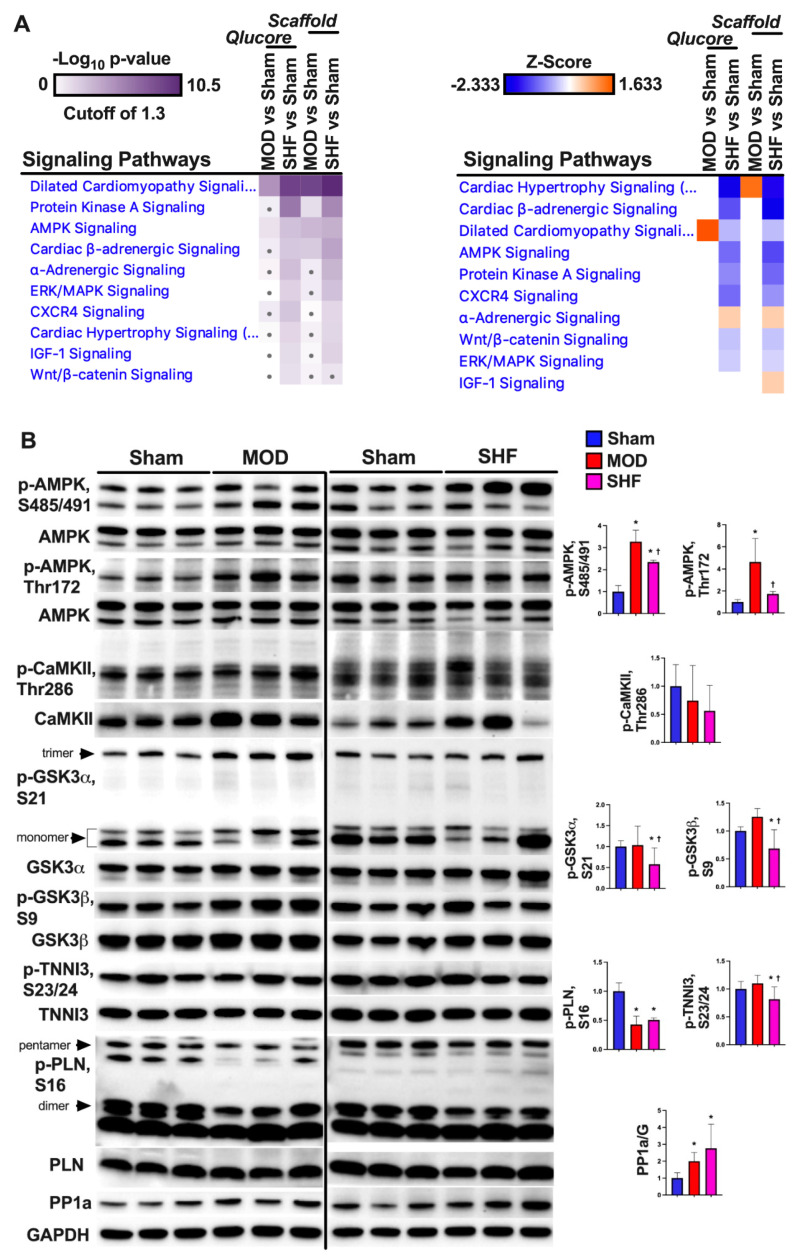
Top-ranked *Signaling Pathways* in the MOD and SHF groups. (**A**) ‘Core Analyses’ of the statistically analyzed p-proteomic datasets by Qlucore and Scaffold bioinformatics software was performed in IPA for each of the two-group comparisons (MOD vs. Sham and SHF vs. Sham), which were then compared in IPA’s ‘Comparison Analyses’ function to yield the most enriched *signaling Pathways* that were shared among the two-group comparisons. The generated heat maps are presented by *p*-value (left) and by z-score (right). (**B**) Western blot showing post-translational modification of AMPK and PKA downstream protein targets as well as protein phosphatase 1 (PP1) expression in the MOD and SHF groups, * *p* < 0.05 vs. Sham and ^†^
*p* < 0.05 vs. MOD.

**Figure 8 ijms-23-00235-f008:**
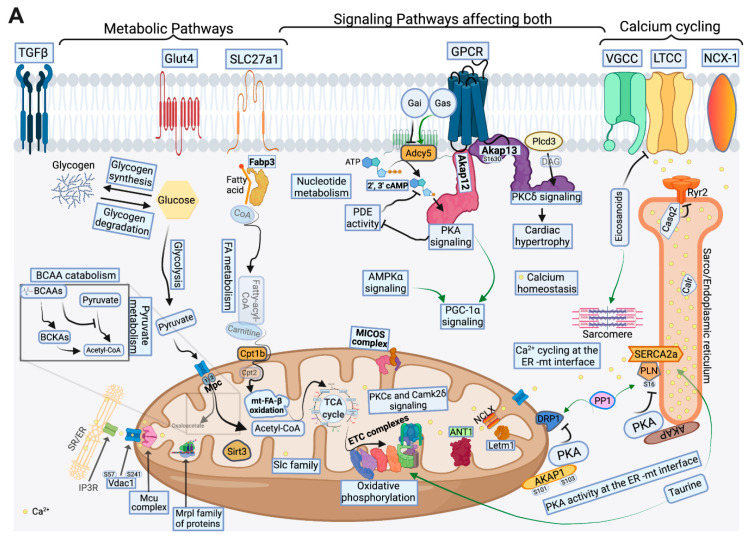
Schematic drawing highlighting metabolic pathways, calcium cycling, and implicated signaling pathways in Sham (**A**), and their directional change in MOD vs. Sham (**B**), and SHF vs. Sham (**C**). Green arrows promote pathway or activity. Rectangles show signaling pathways or proteins that were inhibited or down-regulated (green), respectively, and those that were activated or up-regulated (red), respectively. The deeper the color, the higher the intensity of activation/up-regulation or inhibition/down-regulation, as shown by the scale bar at the bottom of the figure. Oxidative phosphorylation (OXPHOS), fatty acid (FA) metabolism, and mitochondrial (mt)-FA β-oxidation were the most enriched/inhibited metabolic pathways in MOD vs. Sham, followed by BCAA catabolism and enzymes of the tricarboxylic acid (TCA) cycle. Glycolysis and pyruvate metabolism were the least enriched/inhibited in MOD vs. Sham. The metabolites taurine and 2′,3′cAMP were downregulated in MOD vs. Sham. Endoplasmic reticulum (ER)–mt interface protein kinase A (PKA) signaling and calcium cycling as well as PGC-1α signaling were attenuated in MOD vs. Sham. Glycogen degradation and protein kinase C isoform delta (PKCδ) signaling were activated in MOD vs. Sham. In SHF, there was further inhibition or activation of the aforementioned metabolic and signaling pathways, except for glycogen degradation, which decreased in SHF vs. MOD. Unique findings in SHF included decreased PKA activity throughout the cellular compartment, decreased AMPK and mt PKC isoform epsilon (PKCε), and mt calcium/calmodulin kinase 2 delta (Camk2δ) signaling. Derangements in calcium cycling and calcium homeostasis were also evident in SHF. Mitochondrial pathophysiological processes, other than OXPHOS, that are indirectly involved in or regulating OXPHOS (mt calcium uptake and efflux, MICOS complex, mt import and carrier proteins, and mt protein deacetylation) were attenuated in SHF vs. Sham. Additionally, mt translation/elongation was attenuated in SHF. Eicosanoids decreased in abundance in SHF vs. Sham, while 2′,3′cAMP down-trended in SHF vs. MOD. Abbreviations: TGFβ: transforming growth factor beta, Glut4: glucose transporter family 4, Slc27a1: long-chain FA transport protein 1, Fabp3: FA binding protein 3, CoA: Coenzyme A, BCAA: branched-chain amino acids, BCKAs: branched-chain ketoacids, Vdac1: voltage-dependent anion channel isoform 1, Mcu: mt calcium uniporter, Cpt1b: carnitine-O-palmitoyltransferase isoform b, Sirt3: Sirtuin 3, Mrpl: mt 39S ribosomal proteins, Slc25: mt solute carrier family proteins, ANT1: ADP/ATP translocase 1, Letm1: proton/calcium exchanger, NCLX: sodium/calcium/lithium exchanger, PP1: protein phosphatase 1, DRP1: dynamin-related protein 1, AKAP: A-kinase anchor protein, PLN: phospholamban, SERCA2a: Sarco/endoplasmic reticulum calcium ATPase, Calr: calreticulin, Casq2: Calsequestrin-2, Ryr2: ryanodine receptor isoform 2, NCX-1: sodium/calcium exchanger, LTCC: L-type calcium channel, VGCC: voltage-gated calcium channels, GPCR: G protein-coupled receptor, Plcd3: 1-phosphatidylinositol 4,5-bisphosphate phosphodiesterase delta-3, Adcy5: adenylate cyclase type 5, Gai: G protein subunit alpha i2, Gas: GNAS complex locus, and PDE: phosphodiesterase. Note: Both 2′, 3′ cAMP and 3′,5′ cAMP were identified and were considered duplicate metabolites (please refer to methods section for details). They are represented as 2′,3′ cAMP here.

## Data Availability

The corresponding author have all data available upon request.

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
