# Peer review of "Multi-Omics Approach Profiling Metabolic Remodeling in Early Systolic Dysfunction and in Overt Systolic Heart Failure"

_ijms, 2021, doi:10.3390/ijms23010235_

Round 1

Reviewer 1 Report

Study by Chaanin et al. investigated changes in mitochondrial proteome during metabolic remodeling in 2 rat models of heart failure - early systolic dysfunction and overt systolic heart failure. Applying advanced proteomic, phosphoproteome and mebolomic methods they found that mitochondrial proteins involved in respiration and metabolism were downregulated in both models and that more severe changes were observed in the early systolic dysfunction model compered to the overt systolic heart failure. They also showed that PKA and AMPK signaling was changed in these two models. In principle, these findings are novel, interesting and important for understanding molecular mechanisms involved in pathogenesis of heart failure as they provide new evidence supporting the concept that altered mitochondrial function due to changes in mitochondrial proteome are the cause rather than the consequence of heart failure. There are just a few issues that need revision.

  1. In the abstract, it is advisable to avoid un-necessary abbreviations, e.g. mt.
  2. Introduction seems to be too general and not well explaining why it was important to analyze changes in mitochondrial proteome and PKA and AMPK signaling pathways in heart failure.
  3. Concerning terminology, electron transfer chain (ETC) consists of 4 complexes, whereas oxidative phosphorylation system (OXPHOS) involves 5 complexes - I-IV ETC complexes and ATP synthase which is also complex V. This must be corrected thruought the text.
  4. Discussion, lines 461-465: this sentence is rather complicated and difficult to understand, thus must be corrected.
  5. The text should be carefully revised for missing words and other errors. Also check the Reference list (correct: 24. 24…., 27. 27….

Author Response

Study by Chaanine et al. investigated changes in mitochondrial proteome during metabolic remodeling in 2 rat models of heart failure - early systolic dysfunction and overt systolic heart failure. Applying advanced proteomic, phosphoproteome and mebolomic methods they found that mitochondrial proteins involved in respiration and metabolism were downregulated in both models and that more severe changes were observed in the early systolic dysfunction model compered to the overt systolic heart failure. They also showed that PKA and AMPK signaling was changed in these two models. In principle, these findings are novel, interesting and important for understanding molecular mechanisms involved in pathogenesis of heart failure as they provide new evidence supporting the concept that altered mitochondrial function due to changes in mitochondrial proteome are the cause rather than the consequence of heart failure. There are just a few issues that need revision.

In the abstract, it is advisable to avoid un-necessary abbreviations, e.g. mt.

 We have removed unnecessary abbreviations from the abstract as suggested.

Introduction seems to be too general and not well explaining why it was important to analyze changes in mitochondrial proteome and PKA and AMPK signaling pathways in heart failure.

 We have made changes to the last paragraph of the introduction section to address the reviewer’s concerns. Please refer to the tracked version of the manuscript. We hope that our changes are satisfactory to the reviewer in addressing his/her concern.

Concerning terminology, electron transfer chain (ETC) consists of 4 complexes, whereas oxidative phosphorylation system (OXPHOS) involves 5 complexes - I-IV ETC complexes and ATP synthase which is also complex V. This must be corrected thruought the text.

 We have corrected the terminology throughout the manuscript as suggested by the reviewer.

Discussion, lines 461-465: this sentence is rather complicated and difficult to understand, thus must be corrected.

 We thank the reviewer for his/her comment. The sentence was corrected for clarity. It now reads:

“Our findings suggest that mt-pathophysiological changes in MOD are predominantly related to attenuation in mt-FA β-oxidation, BCAA catabolism and OXPHOS. Upon progression to SHF, there is worsening of the aforementioned pathophysiological processes. Additionally, there is impaired pyruvate metabolism, enhanced mt-degradation and apoptosis (impaired mt-calcium regulation and reactive oxygen species [ROS] production)[5, 8, 9, 24, 25], and derangements in the mt-import systems. These mt-pathophysiological changes constitute the drive for the worsening in mt-function and myocardial remodeling, and the transition to an advanced stage of HF.”

The text should be carefully revised for missing words and other errors. Also check the Reference list (correct: 24. 24….,27. 27….

We have proofread the text and corrected any mistakes. Also, suggested corrections to the reference list were made.

Reviewer 2 Report

In this manuscript, the authors analyzed metabolic remodeling in the progression of heart failure by using multi-omics approach. They found that mitochondrial proteins were predominantly down regulated in the transition from early to overt systolic dysfunction, compared with sham, in a rat model of ascending aortic banding model. These findings are compartible with the general concept that oxydation is shifted from fatty acid to glucose in myocardial mitochondria in the development of heart failure. Metabolomic remodeling, compared with the conventioal morphological analysis, is supposed to be a more sensitive tool in the detection of early stage heart failure.

In the animal model of ascending aortic banding, the author differentiate overt systolic heart failure (SHF) from early systolic dysfunction (MOD) by the dilatation of LV (at both end-diastole and end-systole, and thus a drop in ejection fraction),  and incras e of right ventricle weight. However, drop of LVEF and hypertrophy of RV in a pressure-overload model (such as ascending aortic banding or aortic stenosis) is an terminal phenomeum and the advanced myocardial fibrosis may introduce confounding factors in the tissue specimens. This may be the cause of the inconsistency in the attached blots. 

Moreover, a diagram would be helpul to explain the physiological significance of the complex findings bases on the multiple bioinformatic analyses.

Author Response

In this manuscript, the authors analyzed metabolic remodeling in the progression of heart failure by using multi-omics approach.They found that mitochondrial proteins were predominantly downregulated in the transition from early to overt systolic dysfunction, compared with sham, in a rat model of ascending aortic banding model. These findings are compartible with the general concept that oxydationis shifted from fatty acid to glucose in myocardial mitochondria in the development of heart failure. Metabolomic remodeling, compared with the conventioal morphological analysis, is supposed to be a more sensitive tool in the detection of early stage heart failure.

In the animal model of ascending aortic banding, the author differentiate overt systolic heart failure (SHF) from early systolic dysfunction (MOD) by the dilatation of LV (at both end-diastole and end-systole, and thus a drop in ejection fraction), and incras e of right ventricle weight. However, drop of LVEF and hypertrophy of RV in a pressure-overload model (such as ascending aortic banding or aortic stenosis) is an terminal phenomeum and the advanced myocardial fibrosis may introduce confounding factors in the tissue specimens. This may be the cause of the inconsistency in the attached blots.

 We thank the reviewer for his/her comment. This could be part of the reason for inconsistency in the WB as we often see biological variations within the studied experimental groups in terms of response to stressor and disease development. We used criteria published elsewhere (J Vis Exp. 2020 Apr 30;(158). doi: 10.3791/60954.PMID: 32420983) to characterize and phenotype the animals under the proper studied experimental groups.

 Moreover, a diagram would be helpul to explain the physiological significance of the complex findings bases on the multiple bioinformatic analyses.

 Figure 8 is a schematic drawing summarizing important pathophysiological findings in regard to metabolic remodeling in MOD and SHF vs Sham.